

# 1  The effect of propagation saw test geometries on critical cut length

Bastian Bergfeld[1]*, Karl W. Birkeland[2], Valentin Adam[1,3], Philipp L. Rosendahl[3], and Alec van Her-
wijnen[1]
[1] WSL Institute for Snow and Avalanche Research SLF, Davos, Switzerland
[2] Birkeland Snow and Avalanche Scientific, Bozeman, Montana
[3] Institute of Structural Mechanics and Design, Technical University of Darmstadt, Darmstadt, Germany
*Correspondence to*: Bastian Bergfeld (bastian.bergfeld@slf.ch)
**Abstract:**
For a slab avalanche to release, a crack in a weak snow layer beneath a cohesive snow slab has to initiate and propagate.
Information on crack propagation is essential for assessing avalanche triggering potential. In the field, this information
can be gathered with the Propagation Saw Test (PST), a field test that provides valuable data on crack propagation pro-
pensity. The first PSTs were performed about 20 years ago and standards have since been established. However, there are
still differences in how the PST is performed. Standards in North America require the column ends to be cut vertically,
whereas in Europe they are typically cut at a normal angle. In this study, we investigate the effect of these different column
geometries on the critical cut length. To this end, we conducted 27 pairs of PST experiments, each pair consisting of one
PST with slope normal cut ends and one PST with vertical cut ends. Our experiments showed that PSTs with normal cut
ends have up to 50% shorter critical cut lengths, and the difference predominantly depends on the slope angle and slab
thickness. We developed two load-based models to convert critical cut lengths between the test geometries: (i) a uniform
slab model that treats the slab as one uniform layer and (ii) a layered model that accounts for stratification. For validation,
we compare these models with a modern fracture mechanical model. For the rather uniform slabs of our experiments,
both load-based models were in excellent agreement with measured data. For slabs with an artificial layering, the uniform
load-model predictions reveal deviations from the fracture mechanical model whereas the layered model was still in
excellent agreement. This study reveals the influence that the geometry of field tests and the slope angle of the field site
have on test results. It also shows that only accurately prepared field tests can be reliable and therefore meaningful.
However, we provide models to correct for imprecise field test geometry effects on the critical cut length.
KEYWORDS: stability test, Propagation Saw Test, edge effect, failure initiation
**1 Introduction**
Accurate assessment of fracture initiation and crack propagation is essential to evaluate the potential for triggering ava-
lanches (Schweizer et al., 2016). In this context, the Propagation Saw Test (PST) is a field test that provides valuable
insight into the propensity of cracks to propagate (Gauthier and Jamieson, 2006b). In the past 20 years several studies
investigated the influence of PST geometry. They aimed to provide recommendations for the PST column length (Bair
et al., 2014) or looked into the effect of changing slab thicknesses (Simenhois and Birkeland, 2008). It was also reported
that the critical cut length depends on whether the ends of the PSTs are cut slope-normally or vertically (Gaume et al.,
2017). Although PSTs have been used for approximately 20 years and utilized in various studies (Bair et al., 2013;
Bergfeld et al., 2022; Bergfeld et al., 2021; Birkeland et al., 2019; Gauthier and Jamieson, 2008b), the lack of widely



accepted standards hinders its consistent and reproducible application across locations and practitioners. Standards in
North America require the PST column ends to be cut vertically (CAA, 2016; Greene et al., 2022), whereas in Europe
they are typically cut at a normal to the slope (Sigrist and Schweizer, 2007; van Herwijnen et al., 2016).
This methodological difference could possibly explain why previous studies were not conclusive as to whether the crit-
ical cut length decreases (Gaume et al., 2017, slope normal cuts) or increases (Gauthier and Jamieson, 2008a; McClung,
2009, both slope vertical cuts) with increasing slope angle. In both, North America and Europe the weak layer is most
commonly cut upslope, but in rare cases, the weak layer is also cut downslope from the top. Gauthier and Jamieson
(2006a) investigated this difference experimentally and observed no significant dependence of critical cut length on cut-
ting direction. However, they also found that critical cut length does not depend on slope angle. Another contradictory
statement about the cut length to slope angle relationship. However, the geometric and/or methodological differences of
PSTs are likely to affect the results of PSTs (Gaume et al., 2017; Heierli et al., 2008, Supplement Figure S3). Our study
aims to investigate the effect of different column geometries and cutting directions on the critical cut length, a major
structural property. To achieve this, we conducted a series of side-by-side PST experiments with normal and vertical
ends. In addition, we also investigated the influence of cutting direction (upslope or downslope).

The purpose of these experiments was to demonstrate the influence of PST column geometry and cutting direction on the
critical cut length. We also explain where these differences come from and how the stratification of the snowpack influ-
ence these geometric effects. To this end, we developed a uniform- and layered load-based models to convert between
PST geometries. In addition, the developed conversion models were validated against a modern fracture mechanics model
(Rosendahl and Weissgraeber, 2020; Weißgraeber and Rosendahl, 2023).
**2 Methods**
**Field Experiments**
In January and March 2021, we performed field experiments above Davos in the Eastern Swiss Alps, and in Montana,
United States. All field sites were around 2400 m.a.s.l. and PSTs resulted in all possible propagation outcomes (slab
fracture, crack arrest and full propagation). In Davos, we tested a weak layer consisting of surface hoar (grain size: 2-4
mm), while in Montana the weak layer consisted of depth hoar (grain size: 1-4 mm) (Fierz et al., 2008). Slab thickness
ranged from 52 to 96 cm.

In total 27 pairs of PSTs were performed, with each pair consisting of one test using slope normal ends (results with
superscript $X^N$, Figure 1a) and the other with vertical ends (superscript $X^V$, Figure 1b). For 6 pairs we also performed
pairs of PSTs in which the weak layer was cut in upslope as well as in downslope direction ($r_c^{up}$ and $r_c^{down}$ in Figure 1b).



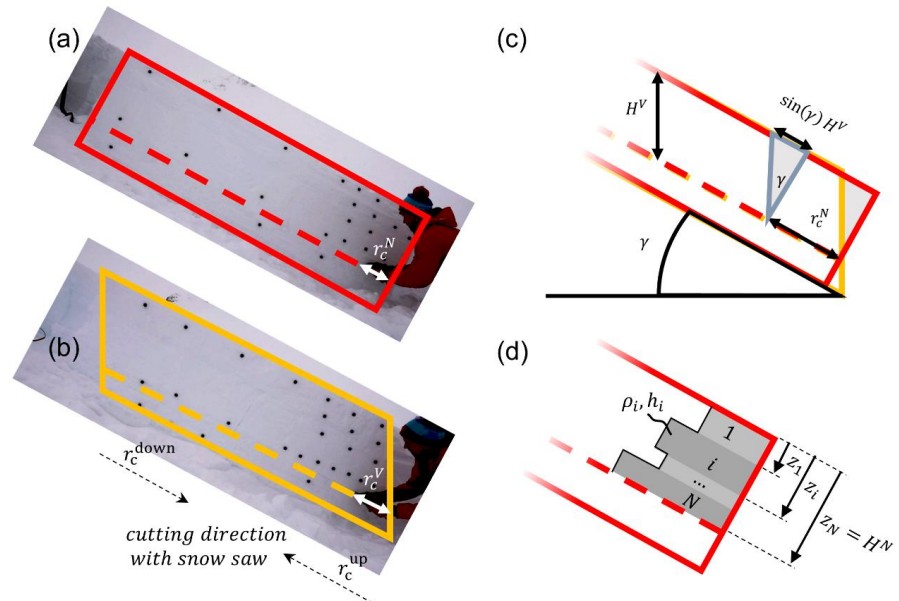

**Figure 1: (a) PST with normal ends and a critical cut length $r_c^N$. The red outline indicates the PST geometry. The dashed line indicates the height of the weak layer. (b) PST with vertical ends and a critical cut length $r_c^V$. (c) Difference in PST geometry. The main difference is the additional slab load for the slope normal geometry shown by the grey triangle. $H^V$ is the vertical measured slab thickness and $\gamma$ the slope angle. (d) In the layered load conversion model, each slab layer $i$ contributes according to their density $\rho_i$, layer thickness $h_i$ and depth in the slab $z_i$.**

For all PSTs, we recorded the critical cut length as $r_c^N$ for PSTs with normal ends, and $r_c^V$ for vertical ends. We then compute the ratio of both cut lengths ${r_c^V}/{r_c^N}$. To investigate the effect of cutting directions, we used the ratio ${r_c^{up}}/{r_c^{down}}$, where $r_c^{up}$ and $r_c^{down}$ indicate whether the critical cut length was taken from upslope or downslope cutting of the weak layer, respectively (Figure 1b).

**Conversion Models**

**Uniform Load Model (ULM).** To find a relationship between $r_c^N$ and $r_c^V$, we assume that the ratio of the cut lengths is inversely proportional to the ratio of the loads of the unsupported slab above the saw cut:

$$\frac{r_c^V}{r_c^N} \propto \frac{\sigma^N}{\sigma^V} = 1 + \frac{\sin(\gamma)\,H^V}{2r_c^N} \qquad (1)$$

If the loads $\sigma^N$ and $\sigma^V$ are expressed through snowpack properties and a uniform slab is assumed, the expression on the right-hand side is obtained, where $H^V$ is the vertically measured slab thickness and $\gamma$ the slope angle. Other snow and geometrical properties (e.g., slab density, beam width, etc.) cancel out. With $H^V = \cos(\gamma)\,H^N$, where $H^N$ is the slope normal slab thickness, equation 1 results in the following model for the conversion of critical cut lengths (assumption of a uniform slab):

$$r_c^V = r_c^N + \frac{\tan(\gamma)\,H^N}{2} \qquad (2)$$



At this point we would like to point out that this relationship (Equation 2) was already suggested in the context of anticrack
nucleation. However, the derivation was based purely on geometric considerations and no further verification was carried
out (Heierli et al., 2008, Supplement Figure S3).
**Layered Load Model (LLM).** The temporal sequence of weather conditions inevitably produces layered slabs in a nat-
ural snowpack. The individual layers differ, among other parameters, in their layer thickness and density. A sloped PST
with layered slab in slope normal geometry results in more (compared to the ULM) load above the saw cut if high density
layers are close to the snow surface (grey triangle in (Figure 1c and d). In addition to the slope angle $\gamma$, the extra load
depends on the individual layer thickness $h_i$, density $\rho_i$, and on the relative depth $z_i$ within the slab (Figure 1d). Concep-
tually, the layered load model is based on the same assumptions as the uniform load model. However, it considers the
layering which makes the formulation to compute the additional load of PSTs with slope normal geometry more intricate:
$$r_c^{\mathrm{V}} = \frac{\sum_{i=1}^{N} r_c^{\mathrm{N}} h_i \rho_i + \frac{\tan(\gamma)}{2} h_i^2 \rho_i + \tan(\gamma) \ (z_N - z_i) \ h_i \rho_i}{\sum_{i=1}^{N} h_i \rho_i} \qquad (3)$$
Where N is the number of layers. For a detailed derivation of the layered load model we refer to Appendix A.

**Layered Mechanical Model (LMM).**
For further verification of the load models, we use a closed-form analytical model for layered snowpacks (Weißgraeber
and Rosendahl, 2023) that was recently validated with field data (Bergfeld et al., 2023), has been utilized. This model
describes the slab as shear-deformable, layered beam, and allows cylindrical bending, while the weak layer is represented
as a layer of smeared springs. We used the model to determine the critical energy release rate G from the measured critical
cut length, depending of the geometric configuration ($G_c^{\mathrm{N}}$ or $G_c^{\mathrm{V}}$, respectively). This critical energy release rate, also called
specific fracture energy, is a material property of the weak layer describing its resistance to crack growth, and it is hence
a proxy for the fundamental physical process of crack growth in PSTs. Subsequently, we used the critical energy release
rate determined from an experiment with slope normal beam ends to calculate back to the critical cut length of a vertically
cut PST. This model is therefore also suitable to convert a critical cut length measured in one PST configuration to another.
Compared to the ULM (Equation 2) and the LLM (Equation 3), the LMM requires many more snowpack properties.
However, it represents the specific snowpack layering of a PST and its influence on the critical cut length in much more
detail, as it takes into account the full deformation behaviour of the slab and weak layer system. We therefore used the
LMM to verify the influence of an asymmetrically layered slab on our load-based models (ULM, LLM).
**Results**
In total we performed 66 PSTs at four different field sites. 54 PSTs aimed to investigate the effect of PST geometry
(Appendix, Table C1), therefore the dataset include 27 pairs of PSTs and each pair consists of one PST with slope normal
and one with vertical PST beam ends. The remaining 12 PSTs were performed to investigate the difference between
upslope- and downslope cutting of a PST (Appendix, Table C2).
**Normal vs. vertical PST ends**
Critical cut lengths were measured between 14 and 70 cm. Overall, $r_c^{\mathrm{V}}$ was systematically larger than $r_c^{\mathrm{N}}$, on average
almost 50 % (colored boxes in Figure 2a).



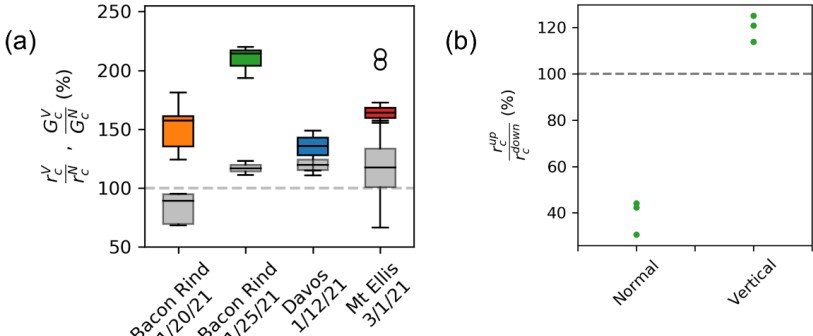

**Figure 2: (a) Ratio of critical cut lengths shown as boxplots for the different field days (colored). Ratio of the critical energy release rates computed with the mechanical model using the critical cut lengths of the experiments (grey). Boxes represent the inter-quartile range with the middle line representing the median value. (b) Ratio of critical cut length from PSTs with downslope and upslope cuts. Results are shown for PSTs with normal and vertical PST ends. Both: The dashed line represents a ratio of 1.**

Differences in snowpack conditions at the various field sites resulted in different deviations between PST geometries.

Median ratios ranged from 136 % to 214 % (Figure 2a, horizontal lines in the colored boxes).

**Upslope vs. Downslope cutting**

Beside PST geometry, the cutting direction also affects the critical cut length. For PSTs with normal ends, $r_c^{up}$ was about

40% of $r_c^{down}$ (Figure 2b, left), while for vertical PST ends $r_c^{up}$ was about 20% longer than $r_c^{down}$ (Figure 2b, right).

Again, these rather large differences can be explained by slab loading and slab mechanics as will be detailed in the discussions section.

**Models**

With Equations (2) and (3) we provide a **uniform load model** and a **layered load model**, respectively. The models allow

us to convert critical cut lengths between the different PST geometries. Our experiments show very good agreement with

both the uniform-load model (Figure 3a, dots) and the layered load model (Figure 3a, crosses). The RMSE between the

measured critical cut lengths in vertical geometry $r_c^V$ and the modelled counterpart is 4.4 cm for the uniform load model

and 4.6 cm for the layered load model.





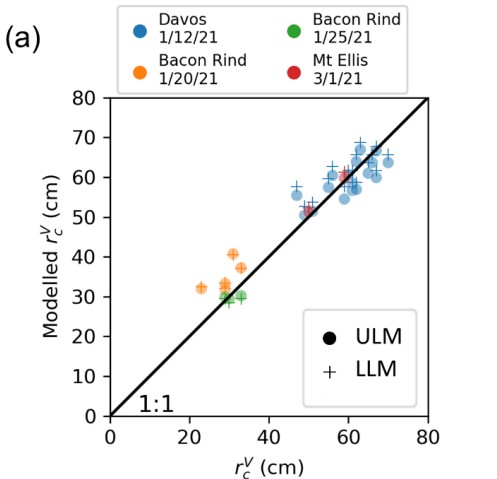
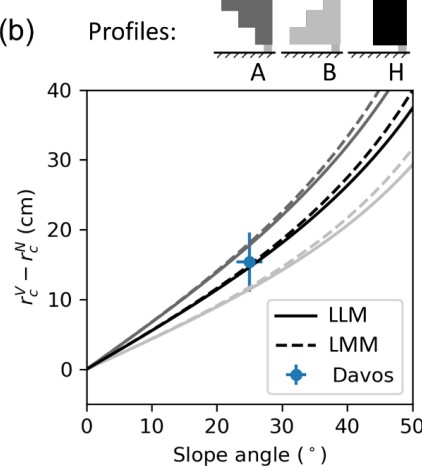

**Figure 3:** (a) Modelled critical cut lengths for upslope cuts with vertical PST geometry $r_c^V$ with the corresponding measured values, dots represent the uniform load model (ULM, Equation 2) and pluses the layered load model (LLD, Equation 3). Different colors indicate the different field days. The black line is the 1:1 line and indicates a perfect model. (b) Modelled differences in critical cut lengths with slope angle. The blue dot represents the mean and uncertainty of the measurements in Davos, as this field day served to define the artificial profiles by matching the mean density. The solid lines are the layered load model and the dashed lines result from the layered mechanical model (LMM). The grey shades indicate different slab profiles given at the top of the figure.

Using the **layered mechanical model** to analyse the global energy balance at the onset of crack growth, we derived critical energy release rates from the experimental data. The model considers the layering and geometrical configuration of a PST experiment to determine the critical energy release rate at the critical cut length, i.e., the specific fracture energy. Unlike the critical cut length, the critical energy release rate is a material property of the weak layer and should thus not depend on test geometry. In fact, the determined critical energy release rates, measured in the different PST configurations (vertical or normal beam ends), differed by a maximum of 20% (Figure 2a, grey boxes), whereas the deviations of the critical cut length were up to six times larger (Figure 2a, coloured boxes).

Our **uniform load model** considers a homogeneous slab and gives a tangential slope dependence (see Equation 2 and black solid line in Figure 3b). For comparison, the **layered load model** and the **layered mechanical model** were evaluated for many different slope angles (Figure 3b, solid and dashed lines, respectively) and 3 different generic slab configurations (Figure 3b, top). In profile H the mean slab density matched the observed snow cover at our experiments in Davos. The direct comparison for the artificial profile H shows a very good agreement between the load models and the mechanical model (compare black solid line and black dashed line in Figure 3b). Note that for profile H the two load models are equal. The deviations of the critical cut lengths ($r_c^V - r_c^N$) measured in Davos can be reproduced very accurately with all models (Figure 3b, black lines and blue dot). In the asymmetric profiles A and B, additional artificial layers with the minimum and maximum density of the Davos snow profile were inserted. For these highly asymmetric slabs (grey lines in Figure 3) there are deviations between the models. Of course, the uniform model cannot represent any differences induced by the layering. However, the layered load model and the mechanical model show good agreement over the entire angle range, whereby the deviations slightly increase with increasing slope angles.



## Discussion

### Normal vs. vertical PST ends

PSTs with slope-normal and vertical ends showed large differences in the measured critical cut length. These differences can be explained with the different PST geometries and the corresponding slab-induced loading of the weak layer. We assume that PST beams were long enough, so that the tail end of the PST beam remains mechanically unchanged when the saw cut is increased and is therefore not relevant (Bair et al., 2014). The constellation is as shown schematically in Figure 1c. Even with no saw cut, the slope normal PST geometry already has an "unsupported" portion of the slab above the weak layer (Figure 4a, blue area at the right beam end). This additional load, in normal geometry, generates higher stresses in the weak layer (and higher energy release rate), leading to shorter critical cut lengths. The shorter critical cut lengths can therefore easily be attributed to this additional load. However, the extent of the difference depends on snowpack properties (e.g. slab thickness, density layering) and slope angle.

This emphasizes that a measured critical cut length can only be interpreted for stability assessment if the applied geometric PST configuration (including slope angle) is considered. In other words, our data show that two equal snowpacks, which should exhibit a similar crack propagation propensity, likely result in completely different critical cut lengths depending on how the PST beam ends were cut and on which slope angle the PST was performed. To ensure comparability of measured critical cut lengths, it is thus imperative to account for the geometrical configuration and snowpack layering, using the models presented.

### Upslope vs. Downslope cutting

When cutting upslope, there is an additional part of the slab that induces an extra load on the weak layer in the normal configuration (Figure 4a, blue area at the right beam end). When cutting from the top, however, a part of the slab is missing, and there is less load (Figure 4a, blue area at the left beam end). The critical cut length of the upslope cut is thus much shorter, in our experiments about 60% shorter (left side in Figure 2b).

In the vertical configuration, on the other hand, the load over the saw cut is always the same, independent of the cutting direction. The observed differences, however, come from the differences in shear stress at the crack tip. Indeed, at the weak layer, there are two shear stress components: (i) shear stress from the slope parallel gravitational pull on the slab (Figure 4b, arrows in the middle), and (ii) bending induced shear stresses (Figure 4b, arrows at the left and right beam end). The slope parallel gravitational pull is always in the same direction (downslope). The bending induced shear stresses at the height of the weak layer, on the other hand, are always in the cut direction. When cutting the weak layer from the bottom upwards, both contributions thus have an opposite effect and partially cancel each other out, while when cutting from the top, both shear stresses have the same sign and add up. This results in longer critical cut lengths when sawing upslope in vertical PSTs. In our measurements, these were 20% longer (right side in Figure 2b).



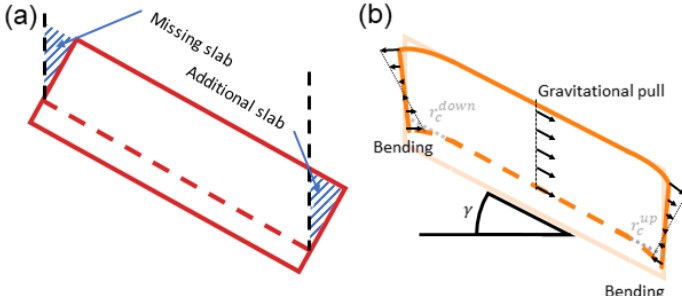

205

**Figure 4: (a) Schematic representation of a PST with normal ends and without a saw cut. The blue marked areas, at the right and left of the PST beam, indicate the additional and missing slab load, respectively, relative to vertical ends (black dashed lines). (b) PST with vertical ends and critical cut lengths $r_c^{up}$ and $r_c^{down}$ for upslope and downslope cutting, respectively. At both PST beam ends the saw cut leads to bending, which results in a stress profile across the slab thickness (black arrows). In the middle part of the PST, the black arrows represent stress in the slab due to the slope parallel gravitational pull. $\gamma$ is the slope angle.**

**Models**

Overall, the load models effectively explained our field results. (Figure 3a). If the RMSE of the uniform load- and layered load model is compared, the uniform load model performs slightly better than the layered load model. However, since our snowpack profiles show relatively homogeneous slabs without pronounced asymmetry (see Appendix D), we would not attach any significance to this minor difference, especially for inhomogeneous and asymmetrical slabs. We believe the layered load model is more accurate. This becomes clear in Figure 3b. Profiles A and B have a density gradient within the slab (asymmetry). Deviations between the uniform and the layered load model seem plausible as higher density layers which are close to the snow surface contribute more to the additional load present in slope normal PSTs (blue hashed in Figure 4a) than if they are deeper in the snowpack. The difference in critical cut lengths is expected to be larger (profile A) or smaller (profile B) than predicted by the uniform load model.

Beside the overall good conversion performance of the models, a systematic offset for PSTs from 20 January 2021 seem to be present (orange dots in Figure 3). We suspect that in these PSTs the beam length was too short, the ratio between slab thickness and beam length was only about 0.5. It is therefore very likely that the geometric difference at the tail end of the beam was also relevant (Bair et al., 2014). However, this is not considered in the models. Overall, our results thus show that the PST geometry plays an important role in the measured critical cut length, and this is mostly driven by differences in load from the slab.

**Model application and limitation:**

PST datasets with different PST configurations can be homogenised using our models. This will increase the comparability and ultimately the scientific utility of these datasets. In addition, it is often the case that the PST ends are cut imprecisely (not perfectly vertical or slope normal) on inclined terrain. The angle of the free edge can easily be determined from photos of the test, and a correction can then be applied using one of the load models with minor modifications (Appendix B). The scatter of the experimentally determined critical cut lengths should thus be reduced.



Beside applications, shortcomings of the suggested load models are evident. The suggested load models rely on Equa-
tion 1, rearranged it reads to: $r_c^V \sigma^V \propto r_c^N \sigma^N$. Using equation A2 (Appendix A) the unit on both sides of the relation is
$J/_{m^2}$, an energy per unit area. The interpretation of Equation 1 is similar to equating both PST configurations (vertical
and normal cut ends) by the specific fracture energy of the weak layer, which is a reasonable approach followed by the
layered mechanical model. However, equation 1 remains freely chosen because it is not the fracture energy that is com-
puted by $r_c^X \sigma^X$. Our experimental results show that the relationship is sufficiently accurate for the conversion of PST
geometries. However, if there are additional changes beyond the PST geometry, the relationship may no longer be suffi-
cient. Imagine additional terms from factors $A$ and contributions $B$ in Equation 1:
$A(\gamma, H^N, \dots) \; r_c^V \sigma^V + B(\gamma, H^N, \dots) \propto A(\tilde{\gamma}, \tilde{H}^N, \dots) \; r_c^N \sigma^N + B(\tilde{\gamma}, \tilde{H}^N, \dots)$
Both can have functional relationships on properties such as slope angle $(\gamma, \tilde{\gamma})$ and slab thickness $(H^N, \tilde{H}^N)$.
As long as such properties remain unchanged $(\gamma = \tilde{\gamma}, H^N = \tilde{H}^N)$, the additional terms cancel each other out and our load
models are applicable.
However, if the critical cut length measured at a certain slope angle and snow cover has to be transferred to a different
situation, the applicability of our models still needs to be confirmed with more experimental work. If necessary, the
functional relationships A and B will probably have to be identified and added. A more generally valid conversion for
critical cut lengths would be of great practical benefit as it allows to extrapolate measured point information on crack
propagation propensity to other slope areas were experimental work is not possible.
**Conclusion and Outlook**
This work has shown that the result of a PST, i.e., the measured critical cut length, is strongly influenced by the test
geometry and cutting direction. PSTs with slope normal beam ends systematically produce shorter critical cut lengths
(48% on average). It also makes a significant difference whether the saw cut in a PST is made in the upslope or downslope
direction (deviations up to 60%). Both deviations can be explained mechanically and are largely controlled by the differ-
ence in slab induced loads. Based on the slab load, a load model was derived for uniform, as well as for layered slabs.
Both models agree well with the experimental results. The comparison with a more sophisticated validated fracture me-
chanical model shows good agreement between all models as long as the slab is largely homogeneous. For layered slabs,
the uniform load model shows greater deviations. The layered load model, on the other hand, shows only minor deviations.
This demonstrates that the fracture mechanical model (LMM) is also largely load-driven in this specific application.
Overall, our results show that the interpretation of measured critical cut length in a PST is not straightforward, as it is
influenced by weak layer properties (specific fracture energy), slab properties (e.g. layering), and test geometry.

Based on our findings, we suggest that PSTs with slope normal ends should be performed with a saw cut in the upslope
direction (Figure 1a). This will ensure that the recorded cut lengths are as short as possible and that the overall test
geometry, particularly the column length, has less influence on the test result. However, this has the disadvantage that the
effects of slope angle are greater than for vertically cut PSTs. In order to compare tests on different slopes, this effect
must be compensated for, which is not straight forward yet. For research purposes, therefore, experiments need to be post-
processed before results from different snow packs, slope inclinations, etc. are compared or combined.



However, if the PST is to be used as a stability tool without further investigation, the vertical PST configuration should
be preferred by practitioners as it allows results to be extrapolated from flatter terrain to steeper slopes with less error.

In general, the use of consistent PST standards will ensure that PST results are easy to interpret, will ensure scientific
rigor and will improve the comparability of tests and their results. In addition, standardization and conversion models
facilitate the comparison of results between researchers, leading to a deeper understanding of snowpack behavior. Prac-
titioners also benefit from standardized methods and interpretation aids that are invaluable in assessing avalanche risk
based on stability tests.

**Appendix A:**
The load above the saw cut of a PST with slope vertical geometry (V-PST) is independent of the slope angle. However,
the load of a PST with slope normal edges (N-PSTS) is not. In sloped terrain, a N-PSTS has more load above the saw cut
than a V-PST. The difference depends on the slope angle, but the layering also has an influence. Layers close to the snow
surface contribute more to the extra load than layers close to the weak layer (of the saw cut). In order to express the
relationship between critical cut lengths $r_c^V$, $r_c^N$ and loads $\sigma^V$, $\sigma^N$ (Equation 1, main manuscript) the loads of layered
snowpacks have to be formulated through density $\rho_i$, thickness $h_i$ and the vertical location $z_i$ of the slab layers $i$ (Figure
A1).

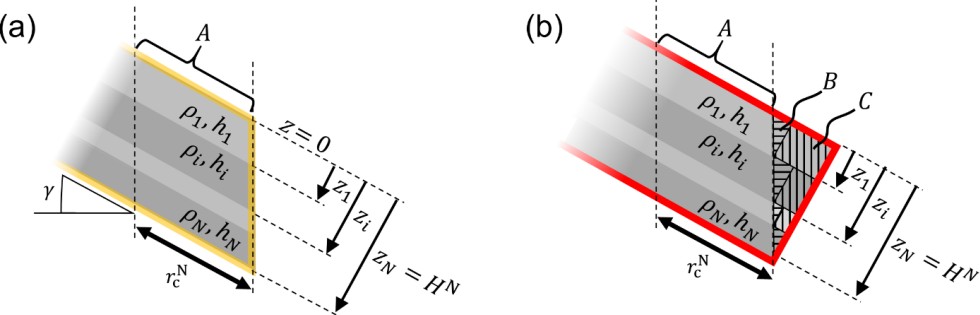


**Figure A1: (a) Schematic representation of a layered slab in a PST with slope vertical geometry (V-PST). (b) PST with slope**
**normal geometry (N-PST). In both cases the load above the saw cut $r_c^X$ depends on the density $\rho_i$ and thickness $h_i$ of the slab**
**layers $i$. In (b), the load of the N-PST depends additionally on the slope angle as the areas B and C increase with increasing**
**angle.**
First for the simpler case of a V-PST (Figure A1a) the mass and load is given by:

$$m_A = r_c^V b \sum_{i=1}^{N} h_i \rho_i \quad (A1)$$

$$\sigma^V = \frac{m_A g}{r_c^V b} = g \sum_{i=1}^{N} h_i \rho_i \quad (A2)$$






Where g is the gravitational acceleration and b is the width of the PST. In the N-PST the Volumes B and C also contribute
to the overall mass located above the saw cut:
$$\sigma^{\mathrm{N}} = \frac{(m_A + m_B + m_C)g}{r_c^{\mathrm{N}} b} \quad (A3)$$

The mass of Volume A remains the same as in Equation A1. The masses $m_B$ and $m_C$ are given by:
$$m_B = \frac{1}{2} h_1^2 \tan(\gamma) b\rho_1 + \frac{1}{2} h_2^2 \tan(\gamma) b\rho_2 + \cdots + \frac{1}{2} h_N^2 \tan(\gamma) b\rho_N = \frac{b \tan(\gamma) \sum_{i=1}^N h_i^2 \rho_i}{2} \quad (A4)$$

$$m_C = (z_N - z_1) \tan(\gamma) h_1 b\rho_1 + (z_N - z_2) \tan(\gamma) h_2 b\rho_2 + \cdots + (z_N - z_N) \tan(\gamma) h_N b\rho_N$$

$$= b\tan(\gamma) \sum_{i=1}^N (z_N - z_i) h_i \rho_i \quad (A5)$$

Putting this together results in the overall mass of:
$$m_A + m_B + m_C = \tan(\gamma) b r_c^{\mathrm{N}} \sum_{i=1}^N \frac{h_i \rho_i}{\tan(\gamma)} + \frac{h_i^2 \rho_i}{2 r_c^{\mathrm{N}}} + \frac{(z_N - z_i) h_i \rho_i}{r_c^{\mathrm{N}}} \quad (A6)$$


Inserting in Equation A3 provides:
$$\sigma^{\mathrm{N}} = \tan(\gamma) g \sum_{i=1}^N \frac{h_i \rho_i}{\tan(\gamma)} + \frac{h_i^2 \rho_i}{2 r_c^{\mathrm{N}}} + \frac{(z_N - z_i) h_i \rho_i}{r_c^{\mathrm{N}}} \quad (A7)$$

As a result, the layered load model provides the relation between the critical cut lengths $r_c^{\mathrm{V}}$ and $r_c^{\mathrm{N}}$:


$$\frac{r_c^{\mathrm{V}}}{r_c^{\mathrm{N}}} \propto \frac{\sigma^{\mathrm{N}}}{\sigma^{\mathrm{V}}} = \frac{\tan(\gamma) \sum_{i=1}^N \frac{h_i \rho_i}{\tan(\gamma)} + \frac{h_i^2 \rho_i}{2 r_c^{\mathrm{N}}} + \frac{(z_N - z_i) h_i \rho_i}{r_c^{\mathrm{N}}}}{\sum_{i=1}^N h_i \rho_i} \quad (A8)$$


$$r_c^{\mathrm{V}} = r_c^{\mathrm{N}} \frac{\tan(\gamma) \sum_{i=1}^N \frac{h_i \rho_i}{\tan(\gamma)} + \frac{h_i^2 \rho_i}{2 r_c^{\mathrm{N}}} + \frac{(z_N - z_i) h_i \rho_i}{r_c^{\mathrm{N}}}}{\sum_{i=1}^N h_i \rho_i}$$

$$= \frac{\sum_{i=1}^N r_c^{\mathrm{N}} h_i \rho_i + \frac{\tan(\gamma)}{2} h_i^2 \rho_i + \tan(\gamma) (z_N - z_i) h_i \rho_i}{\sum_{i=1}^N h_i \rho_i} \quad (A9)$$


**Appendix B:**
The equations derived in appendix A can be used to formulate a model to correct for imprecisely cut PST beam ends. E.g.
the sawing edge of a PST was close to cut slope normal, but with a deviation of angle β from slope normal (or vertical).
As a result, the critical cut length $r_c^{\tilde{N}}$ is measured in such an experiment. To account for this deviation, we have to add a
mass $m_D$ in Equation A6. Note that this "mass" can be negative in the case β is negative (less overhanging mass than the
slope normal cut). The mass $m_D$ has the same contributions as $m_B$ and $m_C$ but is computed from the angle of error β:
$$m_D = \frac{b \tan(\beta) \sum_{i=1}^N h_i^2 \rho_i}{2} + b\tan(\beta) \sum_{i=1}^N (z_N - z_i) h_i \rho_i \quad (B1)$$

At the end, the ratio of the loads and the relation of the cut lengths is given by:





$$\frac{r_c^N}{r_c^{\bar{N}}} \propto \frac{\sigma^{\bar{N}}}{\sigma^N} = \frac{\dfrac{\left(m_A\left(r_c^{\bar{N}}\right) + m_B + m_C + m_D\right)g}{r_c^{\bar{N}} b}}{\dfrac{\left(m_A(r_c^N) + m_B + m_C\right)g}{r_c^N b}} = \frac{r_c^N}{r_c^{\bar{N}}} \; \frac{m_A\left(r_c^{\bar{N}}\right) + m_B + m_C + m_D}{m_A(r_c^N) + m_B + m_C}$$


$$\Rightarrow 1 = \frac{m_A\left(r_c^{\bar{N}}\right) + m_B + m_C + m_D}{m_A(r_c^N) + m_B + m_C}$$

$$\Rightarrow m_A(r_c^N) = m_A\left(r_c^{\bar{N}}\right) + m_D \quad (B2)$$

By inserting the formulations for $m_A$ (equation A1), the formula to correct an imprecisely cut N-PST is derived as:

$$r_c^N = \frac{r_c^{\bar{N}} b \sum_{i=1}^{N} h_i \rho_i + b \tan(\beta) \sum_{i=1}^{N} \dfrac{h_i^2 \rho_i}{2} + (z_N - z_i)\, h_i \rho_i}{b \; \sum_{i=1}^{N} h_i \rho_i}$$

$$= r_c^{\bar{N}} + \frac{\tan(\beta) \sum_{i=1}^{N} \dfrac{h_i^2 \rho_i}{2} + (z_N - z_i)\, h_i \rho_i}{\sum_{i=1}^{N} h_i \rho_i} \quad (B3)$$




**Appendix C:**
**Table C1: Results of 27 pairs of PSTs, critical cut lengths $r_c^V$ and $r_c^N$ indicate whether PST beam ends were cut vertical or slope**
**normal. Slab thickness $H^N$ was measured in slope normal direction. Slope angle is provided in degrees. For further snowpack**
**data we refer to the Appendix D.**

| PST-pairs | Location Date | Critical cut length $r_c^V$ (cm) | Critical cut length $r_c^N$ (cm) | Slab thickness $H^N$ (cm) | Slope angle (°) |
|---|---|---|---|---|---|
| 1 | Davos 1.12.21 | 55 (±2) | 43 (±2) | 62 (±2) | 25 (±2) |
| 2 | Davos 1.12.21 | 49 (±2) | 36 (±2) | 62 (±2) | 25 (±2) |
| 3 | Davos 1.12.21 | 47 (±2) | 41 (±2) | 62 (±2) | 25 (±2) |
| 4 | Davos 1.12.21 | 51 (±2) | 37 (±2) | 62 (±2) | 25 (±2) |
| 5 | Davos 1.12.21 | 56 (±2) | 46 (±2) | 62 (±2) | 25 (±2) |
| 6 | Davos 1.12.21 | 61 (±2) | 45 (±2) | 58 (±2) | 25 (±2) |
| 7 | Davos 1.12.21 | 59 (±2) | 41 (±2) | 58 (±2) | 25 (±2) |
| 8 | Davos 1.12.21 | 65 (±2) | 47 (±2) | 60 (±2) | 25 (±2) |
| 9 | Davos 1.12.21 | 66 (±2) | 49 (±2) | 63 (±2) | 25 (±2) |
| 10 | Davos 1.12.21 | 70 (±2) | 49 (±2) | 63 (±2) | 25 (±2) |
| 11 | Davos 1.12.21 | 61 (±2) | 42 (±2) | 63 (±2) | 25 (±2) |
| 12 | Davos 1.12.21 | 63 (±2) | 52 (±2) | 64 (±2) | 25 (±2) |
| 13 | Davos 1.12.21 | 62 (±2) | 42 (±2) | 64 (±2) | 25 (±2) |
| 14 | Davos 1.12.21 | 62 (±2) | 49 (±2) | 64 (±2) | 25 (±2) |
| 15 | Davos 1.12.21 | 67 (±2) | 45 (±2) | 64 (±2) | 25 (±2) |
| 16 | Davos 1.12.21 | 67 (±2) | 51 (±2) | 67 (±2) | 25 (±2) |
| 17 | Davos 1.12.21 | 60 (±2) | 45 (±2) | 67 (±2) | 25 (±2) |
| 18 | Bacon Rind 1.20.21 | 31 (±2) | 25 (±2) | 57 (±2) | 29 (±2) |
| 19 | Bacon Rind 1.20.21 | 33 (±2) | 21 (±2) | 56 (±2) | 30 (±2) |
| 20 | Bacon Rind 1.20.21 | 29 (±2) | 16 (±2) | 55 (±2) | 30 (±2) |
| 21 | Bacon Rind 1.20.21 | 29 (±2) | 18 (±2) | 55 (±2) | 29 (±2) |
| 22 | Bacon Rind 1.20.21 | 23 (±2) | 17 (±2) | 54 (±2) | 29 (±2) |
| 23 | Bacon Rind 1.25.21 | 29 (±2) | 15 (±2) | 52 (±2) | 30 (±2) |
| 24 | Bacon Rind 1.25.21 | 33 (±2) | 15 (±2) | 53 (±2) | 30 (±2) |
| 25 | Bacon Rind 1.25.21 | 30 (±2) | 14 (±2) | 54 (±2) | 30 (±2) |
| 26 | Mount Ellis 3.1.21 | 59 (±2) | 38 (±2) | 93 (±2) | 25 (±2) |
| 27 | Mount Ellis 3.1.21 | 50 (±2) | 29 (±2) | 95 (±2) | 25 (±2) |







**Table C2: Critical cut lengths measured at Mount Ellis, Critical cut lengths $r_c^{\mathrm{DOWN}}$ and $r_c^{\mathrm{UP}}$ indicate if the weak layer was cut**
**downslope or upslope, respectively. Slab thickness $H^{\mathrm{N}}$ was measured in slope normal direction. Slope angle is provided in de-**
**grees. For further snowpack data we refer to the Appendix D.**

| PST-pairs | Location Date | PST Geometry | Critical cut length $r_c^{\mathrm{DOWN}}$ (cm) | Critical cut length $r_c^{\mathrm{UP}}$ (cm) | Slab thickness $H^{\mathrm{N}}$ (cm) | Slope angle (°) |
|---|---|---|---|---|---|---|
| **1** | Bacon Rind 1.25.21 | Slope normal | 49 (±2) | 15 (±2) | 50 (±2) | 30 (±2) |
| **2** | Bacon Rind 1.25.21 | Vertical | 24 (±2) | 29 (±2) | 52 (±2) | 30 (±2) |
| **3** | Bacon Rind 1.25.21 | Slope normal | 29 (±2) | 33(±2) | 54 (±2) | 30 (±2) |
| **4** | Bacon Rind 1.25.21 | Vertical | 50 (±2) | 50 (±2) | 53 (±2) | 30 (±2) |
| **5** | Bacon Rind 1.25.21 | Slope normal | 33 (±2) | 14(±2) | 53 (±2) | 30 (±2) |
| **6** | Bacon Rind 1.25.21 | Vertical | 24 (±2) | 30 (±2) | 53 (±2) | 31 (±2) |





**Appendix D:**

At each of our four field sites we took a manual profile including density measures. The following four figures are excerpts from the corresponding snow profile databanks.

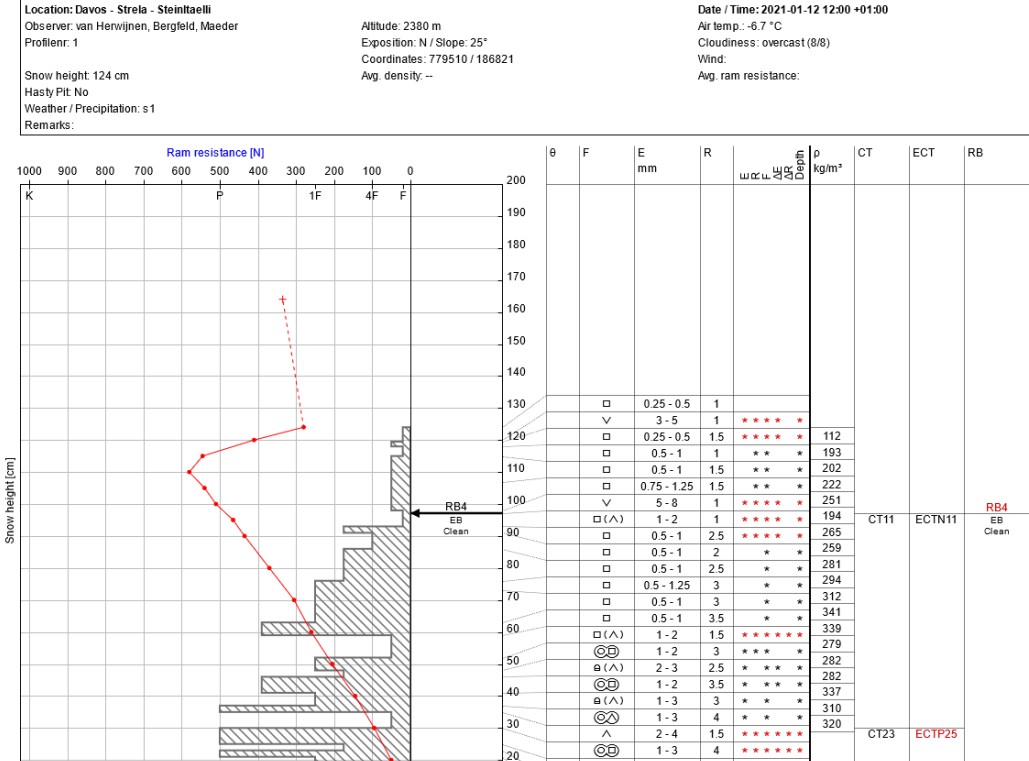

**Figure D1: Manual profile taken at the Davos field site. The hashed area at the left site represents the hand hardness with snow height, the red line snow temperature with snow height. On the right side, grain type, grain size, hand harness, lemons and snow density are given. On the very right, stability test results are written at the height, of the tested weak layer.**



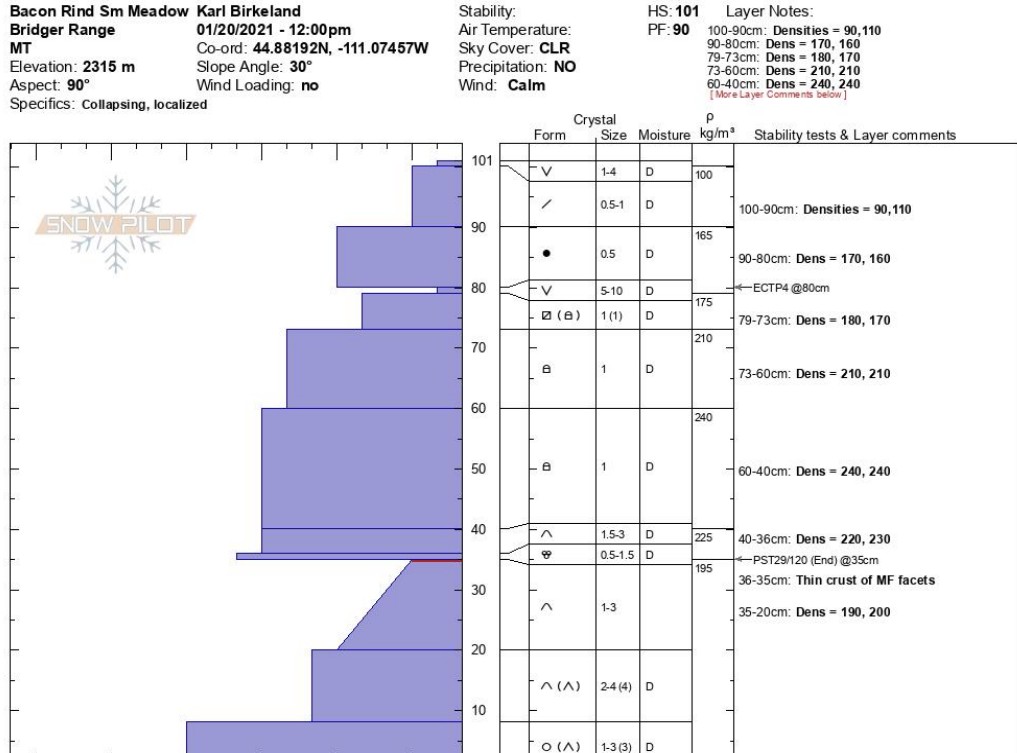

Notes: Pit dug to document snow conditions for research on PST geometry.. Additional Layer Comments: 36-40cm: Dens = 220, 230; 35-36cm: Thin crust of MF facets; 20-35cm: Dens = 190, 200; 20-35cm: Problematic layer;


**Figure D2: Manual profile taken at the Bacon Rind field site on January 20ᵗʰ 2021. The blue area at the left site represents the**
**hand hardness with snow height, On the right side, grain type, grain size, moisture and snow density are given. On the very**
**right, stability test results are written at the height, of the tested weak layer.**



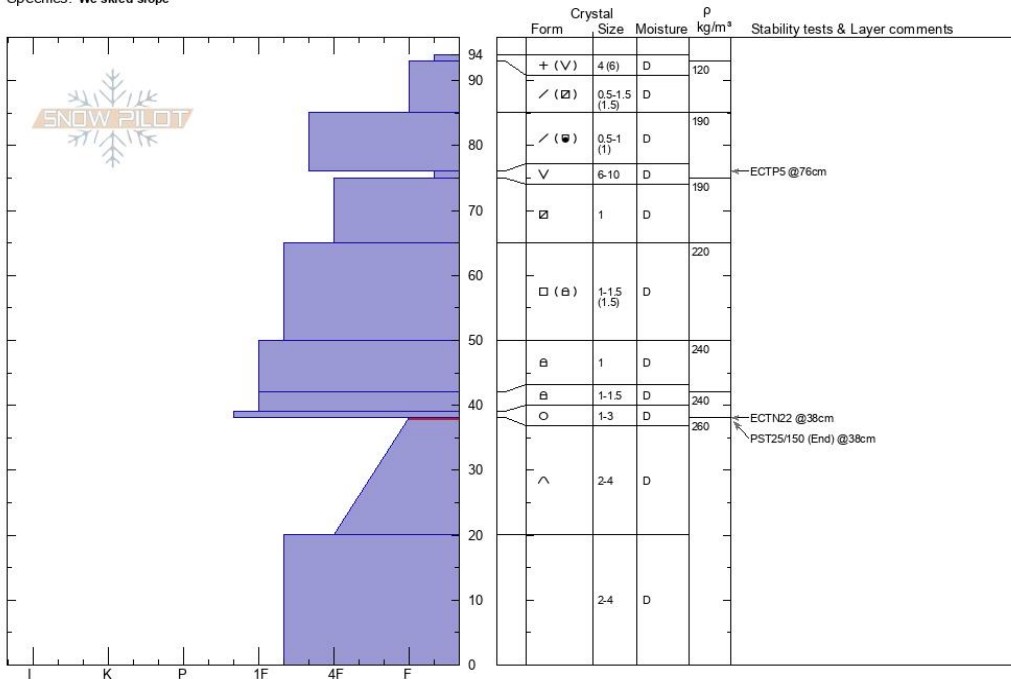


**Figure D3: Manual profile taken at the Bacon Rind field site on January 25th. The blue area at the left site represents the hand hardness with snow height, On the right side, grain type, grain size, moisture and snow density are given. On the very right, stability test results are written at the height, of the tested weak layer.**






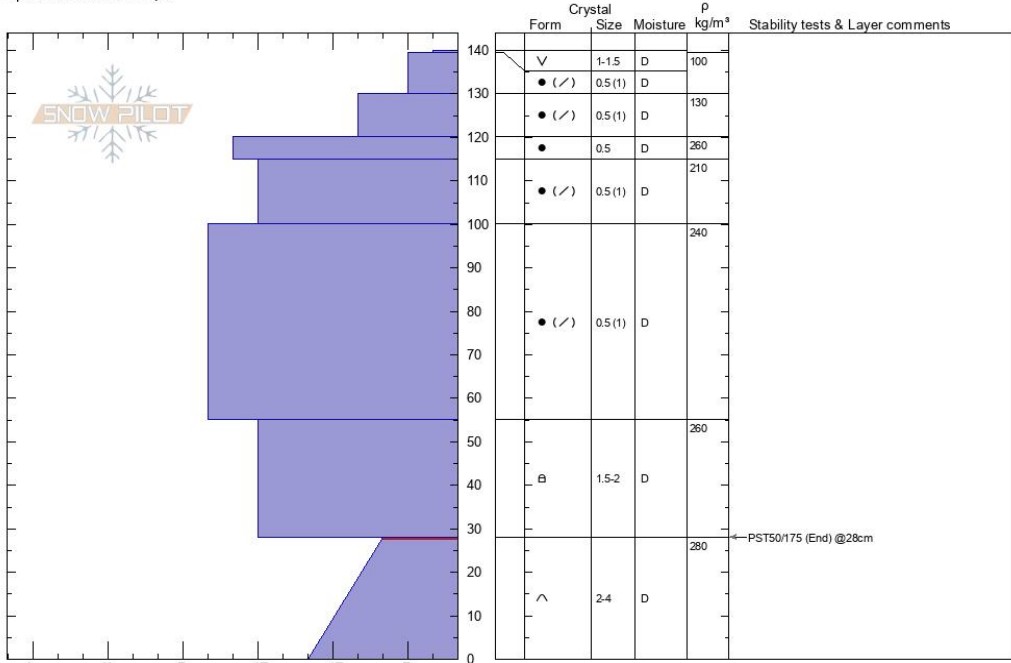

**Figure D4: Manual profile taken at the Mount Ellis field site on January 25th. The blue area at the left site represents the hand hardness with snow height, On the right side, grain type, grain size, moisture and snow density are given. On the very right, stability test results are written at the height, of the tested weak layer.**

## Competing interests

The contact author has declared that none of the authors has any competing interests

## Acknowledgement

We would like to thank Flavia Maeder, Erika Birkeland, and Alex Marienthal for assisting in the field. This research was partly supported by the Swiss National Science Foundation (grant no. 200021_169424) and funded by the Deutsche Forschungsgemeinschaft (DFG, German Research Foundation) under grant no. 460195514.



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
