# Peer review of "The effect of propagation saw test geometries on critical cut length"

_EGUsphere, 2024_

## Author Comment (AC1)

Dear reviewer,

thank you very much for commenting and providing helpful suggestions on the manuscript. Below we have pasted your comments in blue, our point-by-point responses are given in black.

This manuscript provides useful information regarding differences in test protocols related to the "propagation saw test", which is used for snow avalanche stability assessment. There are differences in the test configuration setups that exist between Europe and North America. Field studies carried out in Switzerland and the U.S. compare outcomes associated with the different geometries and two different loading methods are assessed. Accompanying the empirical tests several modeling methods are offered. Notable differences are presented that will be useful in comparing test results or defining a standard.

The paper is well structured, clearly written and offers new relevant results. The title is fitting, and the abstract clear. It is an appropriate topic for NHESS and generally satisfies the criteria for publication in an international journal. In my opinion it will be suitable for publication. That said, I have made some specific recommendations to be addressed to improve the presentation. In particular, it would be beneficial to more robustly present the assumptions made in the mechanical models, as I've noted below.

Thank you for your overall positive feedback and for your careful reviewing of our study.

First, we would like to thank you for your criticism of the assumptions of the mechanical model and explain how we took this into account in the revised version.

Currently, we assumed that the critical cut lengths are inverse proportional to the "load" of the unsupported part of the slab (portion of the slab above the saw cut) $\frac{r_c^V}{r_c^N} \propto \frac{\sigma^N}{\sigma^V}$. As you indicated, this assumption was randomly choosen and not well motivated.

In the revised version we resolved these shortcomings by changing the derivation and motivation of the conversion models. Therefore, we started with a cantilever beam model to explain the mechanical bearing of the slab and used it to derive the conversion model. This way, the conversion models are based on cantilever beam modelling. The new derivation led to equaling the masses of the unsupported portion of the slab for the different PST geometries, which drastically simplified the derivation withouth changing the final models. This will also resolve the issue with the term "load".

In a revised Version of the manuscript we will make the following changes:

- adding Information about new model derivation.
- Modify Equations 1 and 2
- Remove Lines 261 – 266
- Adapt argumentation and equations in Appendix A
- Adapt argumentation and equations in Appendix B

Specific comments:

Line 14 "Standards in North America require the column ends to be cut vertically, whereas in Europe they are typically cut at a normal angle." - As an aside point of curiosity, do you have any knowledge on why (or when) the two different configurations were adopted on the different continents?.
In Europe the PST served mainly as a research tool. The slope normal configuration was (initially) thought to be easier to apply to mechanical crack propagation models (e.g. Heierli). In the US the PST quickly transitioned to a practitioners tool to assess crack propagation propensity. For these purposes, the vertical ends are likely well-suited, as this geometry are less dependent on slope angle. Nowadays, however, I think the argument for one or the other geometry lie mainly in consistency of the datasets.

Line 46 "methodological differences" - What are different methods? Are these inferring formalized differences, or are you referring to unintended variations during the implementation process?

We refer to the methodological differences mentioned above (lines 38 – 46). Namely, the slope normal or vertical cutting of the PST column ends and the upslope or downslope sawing of the weak layer. To avoid misunderstandings, we will rewrite the sentence.

Line 63 Were weak layer thicknesses measured? Was hardness measured (e.g. hand hardness)?

Yes, a complete hand profile was recorded on each field day, in which hand hardness and the complete layering of the snow cover are documented. The profiles are shown in figures D1 to D4 in the Appendix.

Line 66 "For 6 pairs we also performed pairs of PSTs in which the weak layer was cut in upslope as well as in downslope direction" – Suggest changing to "For six additional pairs…". This is clearly presented in the results, but it should be clarified here as well.

Thanks for the suggestion. We will rewrite the sentence.

Line 67 "Figure 1b" - Fig 1b implies that direction is only considered for the slab cut vertically. The up and down superscript notation does not differentiate PST geometry. However, as presented in fig 2b, both N and V were tested.

Indeed, the influence of cutting direction was tested for both PST geometries. We will modify the caption to explain this more clearly, but we will not not illustrate the cutting direction twice.

Suggest that you show, and reference, the upslope and downslope crack length arrows on Fig 1a, and state that the up and down notation applies to both geometries.

See above

Line 71 "(c) Difference in PST geometry ." - add …at the downhill end of the slope normal beam for an upslope saw cut".

We will modify the figure and change the caption.

Line 72 "The main difference is the additional slab load for the slope normal geometry shown by the grey triangle."- Reading this, my initial thought was that there must be a compensating triangle of snow removed from the uphill end of the slab. (You discuss this later when you bring in downslope saw cuts.) When considering the entire beam there would then be no additional resultant slope normal load. What you are referring to is only the portion of the slab directly above the saw cut plus the grey triangle.

Yes, we consider "the load of the unsupported slab above the saw cut". We do not go into more detail in the figure caption as this is explained in the following section: "conversion models", but we will modify Figure 1C and add a missing reference.

Line 78 (Figure 1b) – Same comment as line 67.

See above, we will add a sentence in the caption of Figure 1 to clarify that the cutting direction was determined separately for the different PST geometries.

Line 83 "loads" - These would be more appropriately be defined as stresses, rather than loads. Load is typically used to define a force. The stress in this case is defined as the vertical load acting over the inclined weak layer or saw cut area. As developed in appendix A.

Thanks for this legitimate critisim, with the new model assumptions the model derivation willbased on gravitaitonal body forces, which will resolve this issue.

Line 99 Eq 3 "$rcV$" - Should note that eq 3 reduces to eq 2 for the assumptions stipulated there.

That is correct, we will add a note.

Line 106 "smeared springs"- Define smeared springs. Modeled with shear as well as slope normal elastic properties.

Instead of representing springs as discrete elements, the concept of "smeared springs" involves treating them as a continuous distribution. That means that the modelled weak layer provides a continuous resistance along the column length, rather than at specific points. As you suggest we will add additional information about the elastic properties of the "springs".

Line 178 "This additional load, in normal geometry" what you mean is the additional load above the saw cut area. The total vertical load applied by the beam would be the same. However, it would not be a uniformly distributed vertical load acting over the length of the weak layer + saw cut.

Correct, the total load of the slab remains always the same. It is not dependent on PST geometry, slope angle or cut length. We mean the load induced by the portion of the slab which has no support beneath (blue dashed triangle on the downslope side in Figure 4a). In general, the load models always consider the mass which is not supported by intact weak layer from directly beneath (above/beneath always in gravitational direction). To state this more clearly, we will elaborate on the model derivation.

Line 267 "Based on our findings, we suggest that PSTs with slope normal ends should be performed with a saw cut in the upslope direction" - Here you seem to be suggesting that PSTs with saw cuts from the bottom should in general be the standard.

Line 273 "if the PST is to be used as a stability tool without further investigation, the vertical PST configuration should be preferred by practitioners as it allows results to be extrapolated from flatter terrain to steeper slopes with less error." Here you are suggesting that practitioners should use vertical end cuts.

We see that the last two comments seem to contradict each other. To resolve this, we will revisit the paragraph. In a revised version, we will not make suggestions anymore. We will rather point out the influence on measured critical cut lengths. We think, an understanding on how the slope angle influences measured critical cut lengths is crucial to give recommendations on PST-geometry. However, this study did not investigate the slope angle dependency.

Line 276 "In general, the use of consistent PST standards will ensure that PST results are easy to interpret, will ensure scientific rigor and will improve the comparability of tests and their results. In addition, standardization and conversion models facilitate the comparison of results between researchers, leading to a deeper understanding of snowpack behavior. Practitioners also benefit from standardized methods and interpretation aids that are invaluable in assessing avalanche risk based on stability tests". -Not clear what standard you are suggesting. Possibly two different standards? One for researchers and another for practitioners. Although I have the impression that you are advocating the slope normal for everyone, your intention should be clarified. It seems that the PST may be used more frequently by researchers, than by practitioners for routine assessment since the setup requires a substantial time-consuming effort. If it is to be used as a stability test, practitioners may be interested in assessing the influence of slope angle, in which case might the slope normal configuration have an advantage as a standard? However, this would require using a representative slope that is not in a hazardous area.

As written above we will revisit the paragraph, so that we will not suggest two different standards. We will highlight advantages and disadvantages of the slope-normal PST geometry. As long as there is no model to compensate for slope angle dependence, the slope-vertical geometry seems to be more appropriate as a stability test. If a model becomes available, the "standard" has to be reevaluated.

Line 292 "Figure A1: (a) Schematic representation of a layered slab in a PST with slope vertical geometry (V-PST)." - The saw cut length in the figure, $rc$N, is referencing the N-PST with slope normal geometry instead of $rc$V. As sketched in both Figures A1(a) and (b), A indicates a length equal to the saw cut length.

Thanks for catching this error. Besides, "A" is indicating the Volume of the mass directly above the saw cut, which we will add to the caption.

Line 294 "the areas B and C" - B and C should probably be subscripted with an i to indicate the individual areas. Although below, as in line 303 these are defined or inferred to be volumes identifying the masses mA, mB and mC, which physically is the appropriate designation as applied. This referencing of terms A, B, C as length, area and volume for the same terms needs to be cleaned up for consistency.

In a revised version, the caption will explicitly state that "A" denotes a volume. However, we will not give "B" and "C" the subscript i, as these already indicate the complete volumes. Also, we will not explicitly specify the volumes "B_i" in the equations, so we think the indexing will be confusing.

Line 296 "V-PST (Figure A1a)" - Figure A1(a) shows the saw cut length for the normal beam geometry, although eq A1 is correct.

See above

Line 296 "First for the simpler case of a V-PST (Figure A1a) the mass and load is given by: "- Actually a "stress" acting over the inclined saw cut area, that is in contact with the volume of the slab, A, directly above, as defined by Eq A2. Total vertical load is mA*g.

This is resolved with the new model derivation, See answer on your general remark.

Line 300 "In the N-PST the Volumes B and C also contribute to the overall mass located above the saw cut:" - Assumes the load (force) is determined through the volume defined by the total volume A+B+C acting vertically over the area of the saw cut.

As I interpret it, for both V-PST and N-PST the assumption is that there is no interaction between the isolated snow over the crack and the rest of the slab. Essentially, snow above the saw crack is considered as a free body in which the normal and shear interacting with the rest of the slab are negligible. That is, the rest of the slab is considered independently. Although not explicitly discussed, the "gravitational pull" of the middle part of the PST is presented in figure 4b. However, I don't see how this influences the mechanical model presented. This assumption that the part of the slab over the weak layer can be assumed independent of the rest of the slab should be explicitly presented.

Given a bonded slab here is going to be some interaction at the interface of the slab directly over the saw cut and that over the intact weak layer. While on a level surface it may be slight, intuitively, on a slope this interaction would be exacerbated. Given the different properties of the weak layer and the saw crack area this would, it seems, be particularly evident regarding slope parallel shear.

This will be resolved with the new model derivation, See answer on your general remark. That the slab is not considered will be explicitly stated in a revised version. We will also note that the slab above the weak layer contribute to the overall loading, but these are additive terms which cancel each other as they not depend on PST geometry.

Line 175 "assume that PST beams were long enough, so that the tail end of the PST beam remains mechanically unchanged." – The length of the beam is not relevant in the model. If it is, please explain.

Correct, the model assumes that the edge effect of the far beam end is not relevant.

Line 225 "We suspect that in these PSTs the beam length was too short, the ratio between slab thickness and beam length was only about 0.5. It is therefore very likely that the geometric difference at the tail end of the beam was also relevant (Bair et al., 2014). However, this is not considered in the models." – The ratio of thickness to length is provided. Since the length of the beam does not play a role in the model, a more useful metric may be the thickness.

Here, we give explanation for the systematic offset for PSTs from 20 January 2021. The load models assume that PST column lengths are long enough. However, the experimental dataset involved PSTs with column lengths which were rather short and, therefore, led to deviations of the conversion models (which do not account for column length). That the column length is influencing critical cut lengths for low ratios of thickness to length was already shown (Bair et al, 2014).

The modeled results show good agreement with field test measurements in Figure 3. Accordingly, they are useful to the overall presentation. It is incumbent upon the authors to discuss the lack of importance of the "rest of the beam" in the PST.

The portion of the beam which rests on the intact weak layer contribute to loading as well. However, this loading is uniformly distributed and do not contribute to stress intensification at the crack tip. We will discuss the contribution of the rest of the beam in a revised version.

Line 303 "The mass of Volume A remains the same as in Equation A1." - The mass mA will not be the same in both cases since it depends on the respective saw cut lengths. Should relabel as perhaps mAV and mAN.

That is correct. We will change accordingly.

Line 15 "normal angle" - Perhaps rephrase to "normal to the slope."

Accepted

Line 124 Figure 2 - The two circles that look like 8 in the figure are extraneous. Typo.

No, typically in a boxplot, the individually shown datapoints are outliers. Defined as outside of the range of the whiskers (1.5 times the inter-quartile-range)

Line 252 Suggest that "to extrapolate" - is changed "extrapolation to".

Thanks for the suggestion, we will leave it as is.

Line 253 "were" - should be changed to "where"

Thanks for catching this error.

Line 284 "N-PSTS" - drop the S. "N-PST"

Accepted

---

## Author Comment (AC2)

Dear reviewer,

Thank you for your review and your overall positive feedback on our manuscript. Below we have added your comments in blue and the responses in black.

The paper is an important step towards standardizing techniques for the PST. Standardization bears potential to facilitate and improve future research as it makes results comparable. For practioners the PST entails limitations due to the time consuming execution of the test. However, the findings and analysis on the test provided in this paper do hold great potential in making fracture mechanics and failure initiation more comprehensible in a teaching environment by combining emperical tests with modeling methods and offering mechanical explanations for the results.

Eventhough I believe the manuscript should be accepted as it is, I would like to offer some suggestions:

1. A more precise suggestion on what the findings indicate would be the most suitable standard. It is described that the vertical PST configuration is less susceptible to changes in the slope angle and therefore is suggested to practicioners. To my understanding of the manuscript the benefits also prevail when the crack is initiated from the uphill direction.

In general, we would have liked to have made more precise suggestions in this regard. However, it is not yet possible to say conclusively which PST geometry is best for which applications and situations. Before that, we need to understand more precisely how the slope angle affects test results. Currently, our understanding is not yet sufficient to make a concrete recommendation on PST geometry. We will therefore rewrite the section in the manuscript to discuss the pros and cons rather than making suggestions for specific user groups.

2. A topic that is touched on in the manuscript but not discussed in much detail is beam length. In the models it is assumed that the "beams were long enough, so that the tail end of the PST beam remains mechanically unchanged when the saw cut is increased and is therefore not relevant". In addition, it is mentioned that this did not apply to some results because the ratio between the depth of the weak layer and the beam length was only 0.5. In my opinion it would be interested to discuss this issue in more depth.

We do not elaborate on this, as our study does not provide information on this aspect. Rather, we have explained the observed systematic offset of PSTs results from 20 January 2021. For more information on edge effects in PSTs, we refer to the comprehensive study of Bair et al. 2014, who discussed, among other edge effects, the influencing effect of the far end of the PST column. They concluded that crack propagation was more frequent in shorter tests due to increased stress concentration from the far edge. In their study, this edge effect occurred for PSTs up to 2 m long - or a critical cut length to column length ratio ≥ 0.20. The latter ratio would therefore be more meaningful. In a revised version we will add information about the cut length of beam length ratio.

---

## Author Response (AR2)

Review #1:

Dear reviewer,

thank you again for your careful review of our manuscript. Below we have pasted your comments in blue, our point-by-point responses are given in black. Line numbers refer to the new "track changes file" of the revised Version of the manuscript.

As I stated in my previous review: The paper is well structured, clearly written and offers new relevant results. The title is fitting, and the abstract clear. It is an appropriate topic for NHESS and generally satisfies the criteria for publication in an international journal. Data from the field tests provides new evidence that is significant for both practitioners and researchers. In my opinion it will be suitable for publication following some explanation and clarification. That said, I have made some specific recommendations to be addressed that I hope will improve the presentation.

In this revised version the authors altered the development of the conversion models by introducing a cantilever beam to establish motivation for the development. This offers a reasonable approach. As presented, this resulted in the determination that the masses of the slab above the saw cut are equal for the normal and vertical configurations. The resulting model provided excellent results as demonstrated in figure 3a (for upslope cuts with vertical PST geometry), implying that the mass equivalence assumption may be appropriate. I think that the development equating the masses of the slab above the saw cuts in eq 1 needs to be clarified. While I could be misinterpreting something, I've tried to point out where I have some questions. Since eq 1 is the basis for much of the modeling, I think it is important that the assumptions are clear.

Examining the Conversion Models section:

Line 84: Assumes the cantilever beam does not deform sufficiently to contact the snow beneath the cut.

Indeed, the free hanging cantilever does not deform sufficiently to come into contact with the snow under the cut. We now added this information (line85)

Line 85: "combination of reaction forces…" Would normal force would be more appropriate here? These would all be reactions to the loading.

Thanks for pointing out. We revisited the sentence to be more explicit about the reaction forces. Line (90 – 92)

Line 92-93: "The maximum load a weak layer can support before fracture is reached at the critical cut length. Hence, also $R$ is at a maximum at the critical cut length ($Rmax$)."

Even for a level cantilever beam there is also shear stress to be considered due to the bending moment, as you note in figure 4b. On a slope there will also be a shear as well as a normal component from the weight, also demonstrated in fig 4b. Failure depends on the properties of the weak layer and stress intensity from both the normal and shear stresses at the crack tip. It is likely a mixed mode failure. The ultimate compressive load may not necessarily be reached. For example, surface hoar may be stronger in compression than shear.

Correct, even a leveled cantilever has reaction forces, including a shear component. The picture of a levelled cantilever was drawn for simplicity. However, as we saw that it was misleading we revisited the sentence to not restrict to the flat cantilever anymore. (Line 93)

Line 99: "are independent of PST geometry" As presented in fig 4a, for a cut from the bottom if we were to assume equal critical cut lengths, the total mass, and thus resultant vertical loading above the intact weak layer, are not the same. The vertical configuration would have a greater load. If rN > rV, mass for of vertical configuration is larger, and for rV > rN the mass of the normal may be greater. Perhaps in the model these variations might be considered negligible for sufficiency long slabs. This may coincide with what you are saying on line 195. - 196.

True, edge effects stemming from the far end of the PST column are not considered in the models. In other words, we do not consider differences in the mass of the slab at the far end of the PST column. As elaborated around line 195, this assumption holds as long as the columns are long enough. No changes were made.

Next addressing the rest of the paper sequentially.

Caption in figure 1, lines 76-77: "HV the vertical measured slab thickness" The variable HV is referenced in the figure caption, but it does not explicitly appear in the figure. Nor does it appear explicitly in the paper. However, HN appears in the "Model application and limitation" section" (Lines 261 - 263). It is the same as D in this figure.

Thanks for catching this error. In last revision we changed the Nomenclature from HV and HN to D for the slab thickness.

Line 77"each slab layer i" consider adding, (in both the vertical and normal configurations)

Accepted

Line 126: "G" Should this be subscripted with a c to indicate critical? But you state that it depends on the geometric configuration, so, are you referring to the energy release rate and not the critical? The critical energy release rate is a material property, energy release rate is not.

G_c is correct. We changed.

Line 134: "asymmetrically layered slab" What is the asymmetry to which you are referring? Do you mean layers of different heights and material properties? You should explicitly define what you mean by this at this point in the presentation.

It appears also at lines 236-239.

Line 237: "inhomogeneous and asymmetrical slabs" Clarify the difference.

We elaborate homogeneity and asymmetry in the slab by including the lines 140-142:

*"Uniform slabs or symmetrically (with respect to the centre height of the slab) layered slabs are simplifications, usually slabs have a density gradient so that deeper layers have a higher density and are therefore stiffer. However, the load models take very little account of the effects of asymmetric slab layering. "*

Line 238: "density gradient within the slab (asymmetry)" So by asymmetry you mean density differences of the layers. In general, other properties as well? Which will have relevance for the LMM.

See answer above.

Line 142: "14 and 70 cm" Including both upslope and downslope cuts?

Line 143: "50%" Considering both upslope and downslope cuts, or only upslope? In conclusions (line 273) this is more specific at 48%.

For upslope cutting. We clarified (Line 150).

Line 150: "different deviations" Are you referring to the imprecision in the testing discussed in appendix B. If so, I suggest you express it a more explicitly here. Did you measure a deviation of angle β from slope normal (or vertical)?

No, differences in snowpack conditions (e.g. slab thickness, layering, …) at the various field sites resulted in different deviations between PST geometries. We slightly adapted the sentence to make it clearer( Line 158).

Line 176: "deviations" Which deviations? Same as line 150? Not the same as deviation of angle β from slope normal (or vertical). Line 330?

We deleted "*deviations"* as we saw it is distracting. We just meant differences in measured critical cut lengths due to different PST geometries (vertical vs. normal geometry).

Line 161: "in vertical geometry $r_cV$" Were results similar for the normal geometry?

This is important to note, since differences in geometric configuration is a substantial inquiry of the paper. Does the model apply for both downslope cuts as well as upslope?

We just converted the normal configuration to the (modelled) vertical geometry. Which was then compared to the measured vertical critical cut length. However, Equation 3 can easily be reformulated to do the conversion the other way around ($r_c^N = r_c^V - \frac{\tan(\gamma)D}{2}$). Mathematically, it can also be shown that the RMSE is identical if we model the $r_c^N$ from $r_c^V$.

$$RMSE(r_c^N \ modelled) = RMSE \ (r_c^V \ modelled)$$

$$r_c^N - r_c^V - \frac{\tan(\gamma)\,D}{2})^{\wedge}2 = (r_c^V - r_c^N + \frac{\tan(\gamma)\,D}{2})^{\wedge}2$$

→ True

Does the model apply for both downslope cuts as well as upslope?

In the vertical configuration, as we elaborate in lines 224, the load over the saw cut is always the same, independent of the cutting direction. The load models do not apply. The difference in measured critical cut lengths are attributed to the shear stress at the crack tip (See lines 225-232). For the slope normal PST geometry the load models are able to explain the "greater" observed difference (compared to the difference in vertical geometry) → see Lines 219 – 222. The models, therefore, apply partly.

line 166: "colors" Might add (corresponding to fig 2a)?

As this (now line 175) is the caption of figure 2a, we consider your suggestion as redundant.

Line 261: "factors $A$ and contributions $B$ in Equation 1…" This section is difficult to comprehend until appendix B has been examined. Possibly a short summary at this point would help with the flow of the presentation.

We now provide an example (different from appendix B) of such a "contribution" in lines 267-269.

Line 167: "critical cut lengths" for upslope cuts?

Yes, we added this information (Line 176)

Line 169: "the layered mechanical model (LMM)" I'm not clear on how these critical cut lengths are calculated. Are they derived from previously determined critical energy release rates? But aren't the critical cut lengths to determine the critical energy release rates?

Indeed, the calculation process is outlined in lines 124 – 143. Basically, it is a two-step approach to model $r_c^N$ from a measured $r_c^V$ is:

1. Use $r_c^V$ + "vertical configuration" as input of the LMM to compute the critical energy release rate of the weak layer ($G_c^V$).
2. Use $G_c^V$ + "normal configuration" as input for the LMM to back calculate the modelled $r_c^N$.

Line 262: "slab thickness ($HN, \tilde{H}N$)" HN is equal to D in fig 1d.

We fixed this.

Line 272: "PSTs with slope normal beam ends systematically produce shorter critical cut lengths (48% on average)" Is this considering both cut directions or only upslope?

For upslope cutting. We added the missing information in Line 283.

Line 309: "Volumes B and C" As sketched, B and C are for the top layer. Should indicate that they are meant to be representative of the similar shaped volumes for each i layer.

The Volumes of each layer. In the revised Version we clarified that (Line 321).